# Rational Suicide in Late Life: A Systematic Review of the Literature

**DOI:** 10.3390/medicina55100656

**Published:** 2019-09-29

**Authors:** Carla Gramaglia, Raffaella Calati, Patrizia Zeppegno

**Affiliations:** 1Institute of Psychiatry, Università degli Studi del Piemonte Orientale, 28100 Novara, Italy; patrizia.zeppegno@med.uniupo.it; 2S.C. Psichiatria, Azienda Ospedaliero Universitaria Maggiore della Carità, 28100 Novara, Italy; 3Department of Psychology, University of Milano-Bicocca, 20126 Milan, Italy; raffaella.calati@gmail.com; 4Nîmes University Hospital, 30029 Nîmes, France

**Keywords:** rational suicide, old age, late life, aging, ageism

## Abstract

*Background and Objectives*: The complex concept of rational suicide, defined as a well-thought-out decision to die by an individual who is mentally competent, is even more controversial in the case of older adults. *Materials and Methods*: With the aim of better understanding the concept of rational suicide in older adults, we performed a systematic review of the literature, searching PubMed and Scopus databases and eventually including 23 published studies. *Results*: The main related topics emerging from the papers were: depression, self-determination, mental competence; physicians’ and population’s perspectives; approach to rational suicide; ageism; slippery slope. *Conclusions*: Despite contrasting positions and inconsistencies of the studies, the need to carefully investigate and address the expression of suicidal thoughts in older adults, as well as behaviours suggesting “silent” suicidal attitudes, clearly emerges, even in those situations where there is no diagnosable mental disorder. While premature conclusions about the “rationality” of patients’ decision to die should be avoided, the possibility of rational suicide cannot be precluded.

## 1. Introduction

Suicide is a global phenomenon accounting for 800,000 deaths worldwide every year [1]. Considering the global aging trends in the world and that suicide rates increase with age, suicide in older adults cannot be neglected [2,3,4]. Older adults account for a disproportionately high number of suicide deaths because they are more successful at committing suicide compare to younger adults [5].

In older age there is a high risk of unrecognized and untreated psychiatric illnesses [6,7]. In particular, depression is the most common disorder and the most important risk factor associated with late life suicide [3,8,9,10]. However, most depressed older adults do not become suicidal. Furthermore, approximately 55% of late life suicides are associated with physical illness [3,4], and older people and those with chronic/terminal illness may not have psychiatric comorbidity. Physical illness is more likely to eventually lead to suicidal behaviour when it causes functional disabilities threatening the individual’s independence, autonomy and dignity, quality of and pleasure with life, sense of meaning, usefulness and purpose in life, perceived personal value and self-esteem [11,12,13]. 

Aging may lead individuals to think back about their lives and to experience either feelings of integrity (if they feel their life has been meaningful) or despair (if they are unable to find meaning and achievement in their life) [14]. Moreover, notwithstanding what they have achieved in their life, older adults have to face changes, which are usually losses, in the context of several domains including health, employment, and relationships. Thus, the aging individual is compelled to go through grieving experiences and redefinitions of one’s identity [13] which, if unsuccessful, lead to a damage of the individual’s sense of self. The old age individual’s skills in redefining their physical selves and integrating changes depends on their “historical” attribution of meaning to their body [13,15]. Amery defined aging as an incurable disease, where “the body becomes more and more mass and less and less energy. This mass… is the new, enemy self”, and “alien and authentically adverse self” which is opposed to the “self of the past”[16]: this sense of internal division between one’s sick and healthy selves, with the first being perceived as “alien” by the latter, may lead the healthy self to wish the destruction of the sick one [13,15].

Furthermore, the current historical period is one marked by many new opportunities (including good hygienic conditions, availability of clean water, fresh food, quality public health, progress in medical knowledge and available treatments, starting from antibiotics) which have allowed a rapid increase of life expectancy and longevity, but also by the difficulties and ambiguities in facing what these opportunities have brought about. Life support technologies currently enable the prolongment of critically ill patients’ life, often beyond a point where it can be experienced as meaningful and desirable [17]. Therefore, since living longer does not necessarily mean living a high-quality life, the need to balance benefits and harms of curative and, even more, palliative therapies, especially for painful, terminal illnesses, has opened new topics of legal, moral, and ethical concern.

In recent years, there has been much discussion (not only in the scientific community) about suicide and what has been named “rational suicide” in older adults. It is described as “rational” an action which is “sensible, appropriate, in keeping with one’s fundamental interests, and perhaps even admirable”[18]. Rationality means being capable of deliberating, with no coercion, according to one’s own values and purposes in life [19], and that motives and plans for a certain decision have been thoroughly explored, as well as alternative choices [20]. Therefore, rational suicide may be defined as a sane, well-thought-out and fairly stable decision by an individual who is mentally competent, and who is capable of reasoning and choosing the best alternative among the many available with no ambivalence (see Table 1 for the main definitions and criteria to define the construct of rational suicide).

The concept of rational suicide is controversial and difficult to define, so that some suicidologists and psychologists consider it as an oxymoron. The debate about this topic is even more complicated by emerging and complex to define concepts such as those of “good” or “gentle death”, and of “silent suicide”, defined as the intention to kill oneself by nonviolent means such as refusal of food and liquids or noncompliance with essential medical treatment. Such means of promoting one’s death causes ambiguity in medical, clinical, personal, cultural, ethical, religious and historical interpretations [28]. Moreover, the concept of rational suicide is obviously linked to the debate about the hot topic of euthanasia, physician assisted death (PAD) and physician assisted suicide (PAS) (i.e., if suicide can be rational, then the right-to-die should be legal and regulated).

Given the ongoing debate about the possibility that a “rational suicide” exists, the concern represented by older adults’ suicide and the world population aging trend, we performed a systematic review of the literature about rational suicide in late life, with the primary aim to offer an overview and better understanding of the concept of rational suicide in older adults through a description of the main related topics emerged from the literature.

## 2. Materials and Methods

We searched PubMed and Scopus databases from inception to April 16th and 19th 2019, respectively, with the following search keywords: rational AND suicid* AND (elderly OR older*). Two of the Authors (C.G., R.C.) reviewed and screened eligible articles according to the PRISMA (Preferred Reporting Items for Systematic Reviews and Meta-Analyses) flow diagram [29]. Any disagreement among reviewers was solved through discussion and with the supervision of the third Author (P.Z.). Studies were included if: (1) they focused on rational suicide defined according to one or more of the criteria mentioned in Table 1; (2) they focused on or included/mentioned older adults. To ensure the most complete reporting of the available literature about the topic, we decided to include all available study designs.

Studies were excluded if: (1) they were written in languages other than English; (2) their full-text was not available; (3) they were book chapters, commentaries and editorials/letters to the editor; (4) they were focused on euthanasia, PAD and PAS.

For each study, we extracted: year, design, sample features, used rating scales, reasons mentioned for rational suicide, rational suicide criteria and/or definition, main results, and conclusions.

## 3. Results

### 3.1. Main (Quantitative) Features of the Included Studies

The search retrieved 144 references; 24 full text articles were assessed for eligibility (see Figure 1 for the flow diagram); 1 full text article was not available for the assessment. We eventually included 23 published studies in the review.

The data extracted for each of the studies are described in Table 2, where articles are presented in alphabetical order.

Among the included articles, 6 were original papers [2,30,31,32,33,34], with different designs and aims: 1 was based on coroner records of suicides in older people with and without terminal cancer, and investigated motives for rational suicide [2]; 2 assessed suicidal and non-suicidal older adults, one with a specific focus on self-determination [30] and one on problem solving strategies [31]; 1 surveyed primary care physicians and the impact of patients’ age on their ability in recognizing suicidal symptoms and willingness to treat patients [32]; 1 used qualitative in-depth interviews to assess older adults ideating on self-chosen death [33]; 1 examined older and younger adults’ beliefs and attitudes about late life suicide [37].

Of these six articles, 2 referred to specific criteria or definitions for rational suicide [2,30], and only one out of these two actually used them [2]; in both cases Werth & Cobia’s criteria were mentioned, and Cheung also mentioned Siegel’s criteria; 1 did not mention rating scales [2].

Four out of the six studies explored and mentioned several reasons for rational suicide [2,30,31,33], with the concept of loss being explicitly or implicitly referred to in all these 4 studies. The following topics were mentioned: control, dependence, disability/illness, pain, quality of life (QoL) (3 studies); burden, dignity, loss of pleasure/meaning in life (2 studies) (see Table 2 for more details).

Eleven of the studies described in Table 2 did not include research data, but were specifically focused on old age rational suicide [7,13,25,35,36,38,42,44,45,46,48]: 5 included case(s) descriptions [7,13,25,35,36]; 7 specified the professionals involved in end of life and rational suicide issues.

Criteria for rational suicide and rating scales were mentioned respectively by 3 [13,36,42] and 4 [36,42,44,46] of these studies. Battin’s and Werth/Werth & Cobia were the most frequently reported criteria for rational suicide. Regarding motives for rational suicide, the most frequently mentioned ones were: loss, including loss of meaning and purpose (9 studies); burden (5 studies); control (5 studies); pain (5 studies); QoL (5 studies); autonomy (4 studies); dependence (4 studies) (see Table 2 for more details).

The concepts of ageism and slippery slope effect were discussed by 2 [13,38] and 3 [25,35,44] studies, respectively.

Last, 6 papers were not specifically focused on late life rational suicide but mentioned it [21,28,39,40,43,49]; 3 included case(s) descriptions; all but one[40] specified the professionals involved in end of life issues. Four of these six studies [21,28,39,43] mentioned criteria for the definition of rational suicide: Motto’s [23]; Siegel’s [21] and Werth’s [41]; Werth & Cobia’s [22]; Hoche’s “Balance sheet suicide”[21]. Among the reasons for rational suicide, loss (including loss of meaning) was implicitly or explicitly mentioned by all papers; other reasons included: pain (4 studies), disability (4 studies), control (3 studies), QoL (3 studies), dependence (2 studies), autonomy (1 study) (see Table 2 for more details).

All but one of these six papers [43] mentioned the Hemlock Society and either euthanasia, PAD, PAS, right to die; 2 [21,40] mentioned the Living Will.

### 3.2. Main Topics Covered in the Included Studies

The main topics emerged from the papers included in the review were: depression, self-determination, mental competence; physicians’ and population’s perspectives; approach to rational suicide; ageism; slippery slope.

Furthermore, Table 3 briefly summarizes arguments in favour versus arguments opposing rational suicide emerging from the included studies.

### 3.3. Depression, Self-Determination, and Mental Competence

Though in different ways and from different perspectives, almost all the included articles underscored ambiguities and possible biases regarding old age rational suicide. The point which is most discussed is the possible underestimation of depression in old age patients, either terminally ill or not, and its impact on patients’ self-determination and mental competence. Actually, the current nosology includes suicidal ideation and attempts as symptoms of either major depressive episodes and of borderline personality disorder. Thus, the main emerging and unresolved question is whether suicidal ideation and attempt in late life should be always treated as symptoms of a psychiatric disorder or not.

The results about depression in suicidal older adults in the studies we assessed are mixed; while [2] found that among older patients who died by suicide, those with terminal cancer were less likely than those without to be depressed and to have had previous contact with mental health services [30] reported a greater likelihood of depression in elderly with suicidal ideation than in those with no such ideation. These two studies pointed to the question whether or not underdiagnosis of depression is an issue in older patients’ suicide, and to the importance of assessing self-determination in the context of the rational suicide debate, respectively.

Consistent with the debate about decisions relying both on judgments and affective states [50], poorer problem-solving skills were found in depressed elderly suicide attempters, compared to depressed non-attempters and non-depressed elderly, in contrast with the commonplace belief that suicide attempt in late life would be non-impulsive [31]. Moreover, considering ambivalence, ambiguities and the contrast between rationality and inner uncontrolled compulsion in elderly ideating on self-chosen death, Van Wijngaarden et al. [33] indings seemed to question the concept of rational suicide as an autonomous, free decision without pressure.

Briefly, what emerges from the studies we assessed is that underdiagnosis of depression in old age and especially in terminal illness is a relevant problem [56,57,58] ecause it can be difficult to differentiate depressive disorders from old age-related cognitive impairment and from the normal emotional responses of a person coping with a terminal illness [59,60]. On the other hand, not all suicidal individuals are depressed and mental illness per se does not imply that the individual’s self-determination and competency are compromised. So, even though a careful assessment of every single situation is mandatory, not every suicidal ideation and/or planning should be “unquestioningly” pathologized. Last, an approach focused on the concept of “understandability” of suicidal ideation and/or behaviour has been suggested as potentially more meaningful and useful than that of rationality, because an approach based on rationality might lead to overlooking the expressive and emotional meaning of the wish to die [52].

### 3.4. Views of Rational Suicide: Physicians’ and Population’s Perspectives

Two of the original studies assessed focused on views about rational suicide, on behalf of physicians [32] and of older and younger adults [34].

The first study found that primary care physicians’ skills in recognizing depression and suicidal risk in late life were excellent, nonetheless physicians were less willing to treat older patients, likely due to an age bias leading them to consider suicide intent as rational and normal in that age group [32].

In the general population, a more favorable attitude towards late life suicide was found on behalf of older adults compared to younger adults. When asked about possible protective factors for late life suicide, interestingly, only a small percentage of respondents believed that mental health care could play a preventive role [34]. It is not clear whether this last result reflects mistrust in the possibility of suicidal individual to receive help, or rather the general population’s actual belief that late life suicide is not a clinical issue. Mankind’s history has witnessed changing views about death and suicide, which in clinical contexts have both turned out to be something which should be always prevented, fought and avoided; this is linked to the topic of the principles of autonomy versus beneficence, and of patients’ autonomy and physicians’ responsibility [28,40]. Conflicts inevitably surround the approach to rational suicide also from this standpoint, starting from the ethical principle guiding healthcare professionals, who should respect patients’ autonomy as well as safeguarding their lives. On the other hand, the need of a reflection on the meaning and ethics of care in late life and end of life in the current society has been suggested [28,39].

### 3.5. Approach to Rational Suicide

A careful assessment of patients, a thorough and accurate investigation of the meaning of their attitudes towards end of life, and the development of appropriate supportive strategies to face the new emerging needs in the care of old age patients are described by most of the included papers.

It is clearly underscored that each situation and the actual meaning of the wish to die should be evaluated in the context of individual life experience and histories, which are critical elements for the understanding of rational suicide [2,35,44].

The position described by most of the articles reviewed is an “interlocutory” one, encouraging a thorough exploration of reasons for suicide in the context of a “respectful and humane” relationship between patient and “a trained and insightful professional” [43], based on the belief that most elderly persons “readily discuss their suicidal feelings and intentions and are glad to be able to share their thoughts with someone who cares enough to ask”[61]. Exploration of reasons and meaning of suicidal ideation and/or attempt is particularly important when it clearly emerges a conflictual and ambivalent nature of the decision to die, and when the individual is not realistically appraising her/his problems and prospects [21].

The approach to rational suicide should go beyond the mere assessment of the presence of a mood disorder, and address the whole range of issues possibly contributing to the individual’s wish to die, in order to offer possible targets for interventions, and allow the implementation of approaches aimed at increasing QoL or decreasing pain, distress and suffering, which may eventually lead to a reduction of suicidal ideation [30,42,46,62,63].

### 3.6. Ageism

Albeit sometimes used inappropriately in the existing literature, the term “ageism” means a prejudice, stereotyped assumption, or discrimination made on the grounds of a person’s age, possibly leading to unfair treatment of older ones. From a clinical standpoint, an ageist attitude may lead clinicians to consider older adults’ suicides as rational choices (especially if the person has physical illnesses) and affect clinicians’ ability to identify psychopathology [54]. For instance, depression may seem understandable in the context of an older person’s health and living circumstances, but considering suicidal ideation and wishes to die as understandable just because of the person’s old age is an ageist attitude which should be avoided. Nonetheless, some studies [32] have pointed out that while clinicians’ skills in identifying suicidal behaviour in older adults seem to be preserved, what may be lacking is their willingness to treat patients in this age group, especially if the potential reversibility of their condition is underestimated. This attitude too may reflect ageism and therapeutic nihilism. On the other hand, the implementation of coercive life-prolonging measures with a poor consideration of the individual’s autonomy is another potential form of ageism hidden behind medical paternalism [36].

Trying to find a balance between patients’ autonomy and clinicians’ responsibility, therapeutic nihilism and medical paternalism is necessary to face a phenomenon, which may be associated with increased longevity.

### 3.7. Slippery Slope

Several authors have expressed concern that recognizing the right to die would too easily shift to “a climate enforcing a social, obligatory duty to die”, and novelists have well described this risk in science fiction stories such us Richard Matheson’s The Test (1958).

Indeed, several of the selected papers mentioned the “slippery slope” argument, which is also debated in the end-of-life/euthanasia/PAS/PAD context and which refers to the possibility that acknowledging the right to die would eventually shift towards an obligation to die and/or lead older persons to feel guilty if they had no wish to end their life and decided not to commit suicide [21].

However, it has to be underscored that, in the context of the euthanasia debate, no consensus has yet been reached about the presence of the slippery slope effect [64,65].

## 4. Discussion

The aim of this review was to offer an overview and a better understanding of the concept of rational suicide in older adults.

Opinions are not consistent about whether older adults’ wish to hasten death in the absence of a psychiatric disorder could be regarded as a rational choice. Suicide presenting in the absence of a clinically diagnosable depression (or of other conditions impairing mental function) and occurring in cases of adverse conditions such as physical illness and aging has been considered “rational” from the perspective of respect for individual autonomy, and the rational assessment of utility [52]. Suicide in these situations may be seen as a possibility to regain and exert control and autonomy in the face of a miserable existential condition marked by pain and suffering, before the progressive worsening of one’s physical condition, thus finding an escape from a life which is no longer considered as such or worth living.

Even though Authors do not agree on the role of mental disorders in “rational suicide”, usually those who adhere to the conventional psychiatric opinion think that mental disorders are preeminent with regard to suicidal ideation and that suicide is a manifestation of psychiatric illness, hence they do not consider suicide as a rational option [40,54]. The possibility to disentangle depression and suicidal ideation in old-age suicidal individuals would allow the identification of potential targets for ad hoc interventions, going beyond those specifically directed towards the treatment of depression. For instance, there might still be chances, even for terminally ill and/or disabled older adults, to find relief from their pain and suffering, which could eventually lead to a decrease in suicidal ideation and suicide risk [19]. Furthermore, the perspective considering suicide as a symptom of psychiatric illness overlooks the possibility that there may be circumstances in which suicide or the refusal of life-sustaining medical treatment may result from the rational decisions of autonomous individuals, and may lead to interventions aimed at preserving life which can sometimes be applied at the expense of individual autonomy[40].

Although life expectancy for those living in western societies is higher than ever before in human history, this has had the effect of making it difficult for medical professionals to find a balance between patients’ autonomy and clinicians’ responsibility, to admit when someone is dying and to know where to draw the line in terms of offering more treatment [66,67]. The improvement of late life palliative care and QoL should be possibly accompanied by changes in the medical organization and culture to better address the specific end of life related needs in old age.

Some valuable suggestions to face the problem of rational suicide emerge from the studies we assessed; for instance, Richman, quoted by Moore [44] underscored the primary importance, when meeting a suicidal individual, “to make contact rather than obsess over whether they have a right to live or die”. The importance of an interlocutory and reflective attitude is described by Ruckenbauer and coworkers [48], who state that while suicidal individuals cannot be deprived of their autonomy, nonetheless they should not be fully and without protection be released into their autonomy. Moreover, they underscore that patients cannot be reduced to their medical or psychiatric illness, but at the same time, an overestimation of existential stock-taking should be avoided.

It should be underscored that a limitation of this work is that, since the available literature was very uneven and, despite the selection criteria, mainly based on opinions and theoretical papers, the reported quantitative synthesis is limited. Furthermore, we cannot exclude that the choice of not including editorials and book chapters in the current review might have limited the access to potentially relevant material. Further original research studies may help to better understand this complex topic.

## 5. Conclusions

In conclusion, despite contrasting positions and inconsistencies of the studies described in this review, what seems to emerge is that the expression of suicidal thoughts in older adults, as well as behaviours suggesting “silent” or indirect suicidal attitudes [68,69], should be carefully investigated and addressed, even in the absence of a diagnosable mental disorder. Clinicians should try to decode the possible communicative role of suicidal behaviour while avoiding premature conclusions about the “rationality” of patients’ decision to die, and considering it by default as “reasoned behavioural expression of legitimate preference for an earlier death” [7,46,49,70]. Nonetheless, the possibility of rational suicide cannot be precluded.

## Figures and Tables

**Figure 1 medicina-55-00656-f001:**
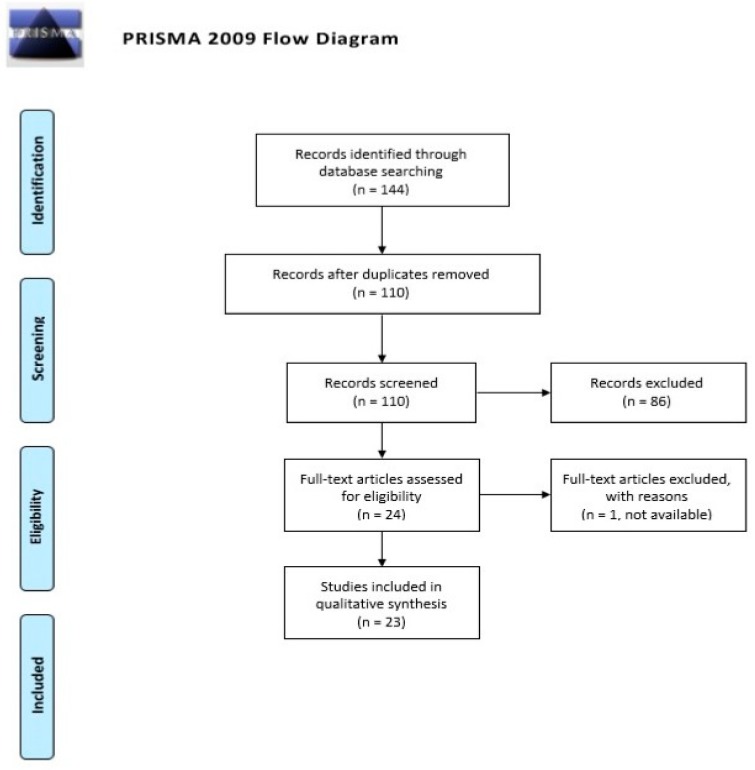
PRISMA 2009 Flow Diagram [29].

**Table 1 medicina-55-00656-t001:** Definitions and criteria for rational suicide.

	Definition/Criteria of Rational Suicide
Siegel [21]	Realistic assessment of the situation on behalf of a person whose mental processes are not impaired by either psychological illness or severe emotional distress. The motivational basis for the decision could be understandable on behalf of uninvolved observers.
Cheung et al. [2]	Add further details about Siegel criterion (3): (1) The person understood the terminal nature of her/his condition. (2) The person consciously disengaged from treatment. (3) The person communicated the desire or made preparations to end her/his life. (4) A triggering event heightened a hopeless situation.
Werth & Cobia [22]	Presence of an unremitting hopeless condition (such as terminal illness, severe pain, both physical and psychological, deteriorating conditions, no longer acceptable quality of life, etc). The decision is a free choice. The decision results from a sound decision making process including the consultation with a mental health professional, and with objective and significant others; a non-impulsive assessment of alternatives, of the possible impact of the decision on significant others, and of the congruence of the decision with the individual’s personal values.
Valente & Trainor [19]	Rational decisions reflect careful planning and consideration of adequate information (e.g., complete and accurate medical facts); preparations (e.g., wills, funeral arrangements); consideration of effect on others, treatment options and alternatives.
Motto [23]	A rational decision should be realistic (i.e., should be made after a realistic assessment of the individual’s situation and after gaining full knowledge of options and consequences) and have minimal ambivalence (i.e., a decision should not be made on the basis of a transient desire and should not be inconsistent with the individual’s longstanding and fundamental values).
Diekstra [24]	Enduring wish to die in a person with a condition of enduring unbearable physical and/or emotional pain, no hope for improvement. The person, who is not mentally disturbed, makes a free will decision which would not cause “unnecessary or preventable harm” to others.
Humphry [25]	“Considered decision” on behalf of a mature adult individual, after reasonable medical help has been sought and the treating physician has been informed. A will should be made and a note should be left. The suicide should not involve others criminally.
Weber [26]	Two meanings of right to die: - right to refuse life-sustaining treatment; - an “affirmative right to obtain death-a right to suicide”.
Graber [27]	A reasonable appraisal of the situation reveals that one would be really better off dead.

**Table 2 medicina-55-00656-t002:** Summary of the included articles in alphabetical order.

				**Empirical Studies Including Data**		
**Author, Year**	**Perspective/Approach**	**Sample Features/Specialists**	**Rating Scales**	**Topics Covered/Reasons Mentioned for RS**	**RS Criteria and/or Definition**	**Main Results and Conclusions**
Cheung et al. 2017 [2]	Focus on the comparison between older people with and without terminal cancer who died by suicide, and analysis of motives for suicide. Hypothesis: in older people with terminal cancer, suicide can be considered a rational choice rather than the result of depression.	Source: Coroner records about suicides in ≥ 65-year-old 07/2007–12/2012, with available data about terminal cancer, N = 214, 74.3% males, 60.7% aged 65–79-year-old N = 23 (10.7%) terminal cancer.	n.a.	- Burden - Control (loss of) - Dependence - Dignity - Functional disability - Pain - Physical illness - Pleasure with life (loss of) - QoL - Sense of usefulness, purpose, value (loss of)	Siegel, 1986 [17] (3rd characteristic: a motivational basis that would be understandable to uninvolved observers) Werth & Cobia, 1995 [22]	- Older patients with terminal cancer who died by suicide were less likely than those without cancer to be depressed and to have had previous contact with mental health services. - 82.6% of terminal cancer cases had a motivational basis for suicide, understandable to uninvolved observers, due to physical (pain, functional disability) or psychological suffering. - Underdiagnosis of depression in patients with terminal cancer? - Choice to end one’s life as a rational act to alleviate suffering?
Fortin et al. 2001 [30]	Focus on suicide prevention. Health care personnel (not mental health professionals) performing assessment.	N = 66 French-Caucasian older adults (age range between 69 and 96 y.o.) with no cognitive deficit, from 7 long-term facilities. N = 11 suicidal (N = 7 males). N = 55 non suicidal (N = 22 matched for age, gender, civil status).	PAQ GDS	- Control (loss of) - Decreased self-esteem - Helplessness - Hopelessness - Losses (physical, psychological, emotional, social, environmental) - Pain - QoL - Relationship problems - Satisfaction	Werth & Cobia (1995) [22], mentioned (but not used)	- No difference in self-determination between older adults with/without SI. - Differences on social subscale: SI have less consideration of own behaviour’s impact on others and less satisfaction with relations with children and family relationships. - SI more depressed than non-SI. - Debate on RS warrants research on self-determination.
Gibbs et al. 2009 [31]	Focus on problem solving strategies, closely related to the topic of suicidality in old age.	N = 64 > 60-year-old MMSE >18 - N = 18 Depressed elderly with past SA. - N = 27 Depressed elderly never suicidal. - N = 19 Non-depressed elderly.	SPSI-R MMSE HAM-D BHS B-SIS SIS B-SLS CIRS-G	- Dependence - Loss - Physical illness	n.s.	- Depressed elderly SA perceived problem solving as dysfunctional and deficient compared to depressed non-attempters and non-depressed elderly: problems perceived more negatively and approached more impulsively and carelessly. This is in contrast with the common clinical view of late-life SA and those who die by suicide as being non-impulsive. - Both depressed groups compared to non-depressed elderly had lower rational and positive problem solving. - Depressed SA showed higher avoidant style than non-depressed elderly. - Lifetime diagnosis of SUD predicted lower total problem-solving score, higher negative problem orientation/impulsivity, and avoidance scores.
Uncapher and Arean 2000 [32]	To determine the influence of patients’ age on primary care physician recognition of suicidal symptoms and the willingness to treat the suicidal patient.	N = 342 physicians (63% response rate), of whom N = 215 primary care physicians, asked to assess 2 vignettes of depressed suicidal patient, either geriatric, retired, age 78 y.o. (N = 100), or young, employed, 38 y.o. (N = 115).	21-item Suicidal Patient Treatment Scale	n.s.	n.s.	- Physicians recognized depression (99%) and suicidal risk (94%). - Physicians were less willing to treat the older patient, feeling that his SI was rational and normal. - Possible age bias?
Van Wijngaarden et al. 2016 [33]	Qualitative in-depth interview study aimed at a phenomenological characterization of the phenomenon “life is completed and no longer worth living”.	N = 25, > 82 y.o., N = 11 males form the Netherlands, ideating on self-chosen death. Inclusion criteria: (1) considered their lives to be ‘completed’; (2) suffered from the prospect of living on; (3) current wish to die; (4) 70 y.o. or older; (5) not terminally ill; (6) considered themselves to be mentally competent; (7) considered their death wish reasonable. N = 23 members of RTD organizations.	Interview HADS	- Burden - Control (loss of) - Dependence - Dignity (loss of) - Interpersonal theory of suicide: thwarted belongingness, perceived burdensomness - Loneliness - Meaning - Pain - QoL	n.s.	- Themes: detachment & attachment; rational & non-rational considerations; taking control & lingering uncertainty; resisting interference & longing for support; legitimacy & illegitimacy. - Rationality versus inner uncontrolled compulsion. - Ambiguities and ambivalence present after a putatively rational decision: need to develop careful policy and support for older people. - Results question the concept of rational suicide as an autonomous, free decision without pressure.
Winterrowd et al. 2017 [34]	To examine beliefs/opinions (most likely precipitants and protectors) and attitudes about older adults’ suicide, in a cultural perspective.	N = 255 older adults (86% European American), 70.95 y.o. mean age, 38% males. N = 281 younger adults (81% European American), 19.04 y.o. mean age, 30% males.	Ad hoc attitudes Scale Personal Attribute Questionnaire-Short Form	n.s.	n.s.	- Precipitants: health problems, mostly in older adults; rational/courageous suicide, admissible (56.7%). - Most favourable attitude about older adult suicide: older adults, persons with more education, persons not identifying with a religion, persons with a history of suicidality. - Older adults suicide viewed as more admissible by males and with more sympathy by females. - Protectors: religiosity in older adults (21.1%); supportive relationships (37.9%) in younger adults. - Mental health care believed to play a preventative role by 6.7% respondents.
				**Case Studies Specifically Focused on Late Life Rational Suicide**		
**Author, Year**	**Perspective/Approach**	**Sample Features/Specialists**	**Rating Scales**	**Topics Covered/Reasons Mentioned for RS**	**RS Criteria and/or Definition**	**Main Results and Conclusions**
Balasubramaniam 2018 [13]	Case presentation: 72 y.o. male, retired, widowed, with adenocarcinoma.	Geriatricians Psychogeriatricians	MCAS (used) DDRS and SATHD (mentioned)	Ageism Control (loss of) Dependence Frailty Gerontophobia Loss QoL Sense of identity/Sense of self (loss of)	n.s.	- Geriatricians increasingly encounter older adults expressing the desire to end their lives, who may have medical illnesses (not necessarily terminal ones), but no diagnosable mental illness. - Is the absence of a diagnostic category to describe a mental state in which suicide appears like the best option a flaw in nosology? - Is RS a rational entity that will be increasingly encountered as views about health, choice, and control continue to evolve?
Lerner 1995 [35]	Case Story of a Couple and Review of Humphry’s Book Final Exit.	n.s.	n.a.	Autonomy (loss of) Terminal illness Terminal old age	n.s.	- Ageism and slippery slope. - Ambiguities surrounding elder suicide. - Need to approach elder suicide in the context of individual life experiences.
Simon 1989 [7]	Clinical/legal issues of silent S + 2 cases (clinical) + case law examples.	Clinicians	n.a.	Autonomy (loss of) Factors including psychological, social, ethical, cultural, economic and situational Losses Medical complaints	n.s.	- Silent S: by non-violent means as self-starvation or non-compliance with essential medical treatments. - Frequently unrecognized because of underdiagnosed depression and/or interjection of personal belief systems of healthcare providers and/or family members. - Cognitive and affective aspects of decision making. - Mental competency impaired (de jure or de facto) by depression: “Premature conclusions that the patient has made a ‘rational’ decision to die must be avoided”; anyway, “certainly every elderly patient who is depressed is not incompetent”. - Treatment: ECT, antidepressants, psychotherapy.
Wand et al. 2016 [36]	2 cases discussed in the light of the importance of a narrative and bio-psycho-social approach to the management of the wish to die.	Psychiatrists Psychogeriatricians	MCAT	Autonomy (loss of) Burden Control (loss of) Coping strategies Dependence Disability External support Helplessness Hopelessness Loss of purpose/meaning/role in life QoL Scared of institutional care Tiredness of living	Battin 1984 [37] Conwell & Caine, 1991 [38]	- Open question about the possible differences between people expressing a wish to die and SA. Does a continuum exist, from wish to die, to SA, to S? - Rationality is probably dimensional rather than dichotomous (Conwell & Caine, 1991). - Requests for euthanasia may occur in older people in the absence of a significant mood disorder. - Ageism and medical paternalism. - Relevant topics: Narrative formulation: Crafting an advance care directive; Exploration of spiritual issues; empathic ongoing care/support; Support both for patients and families; Social interventions.
				**Case Studies Not Specifically Focused on Late Life Rational Suicide (But Mentioning It in the Text)**		
**Author, Year**	**Perspective/Approach**	**Sample Features/ Specialists**	**Rating Scales**	**Topics Covered/Reasons Mentioned for RS**	**RS Criteria and/or Definition**	**Main Results and Conclusions**
Fontana 2002 [39]	Historical and philosophical perspective + case description. RTD, PAS, euthanasia mentioned; Hemlock society mentioned.	Nurses	n.a.	Autonomy (loss of) Control (loss of) Dignity Loss of meaning/purpose in life Pain QoL Self-determination	Siegel, 1986 [17] Werth, 1995 [22]	- Good death as a right. - Problem of having no position and no guiding principle from AMA and ANA. - Implications for the meaning of care (in nursing).
Karlinsky et al. 1988 [40]	Psychological, ethical, legal issues + 2 cases, one advanced age, one terminal illness; euthanasia mentioned; Hemlock Society mentioned.	n.s.	n.a.	Competency Control (loss of) Dependence Disability, physical Mental state/psychiatric illness QoL	n.s.	- Contradictions between the principles of patients’ autonomy and physicians’ responsibility. - Living will.
Rich 2004 [28]	Historical, ethical + case description of chronic AIDS; PAS, euthanasia mentioned; Hemlock Society mentioned; VSED and terminal sedation mentioned.	Nurses	n.a.	Escape from life Terminal illness Disability, permanent Autonomy (loss of) Pain	Siegel, 1986 [17] Werth, 1999 [41] Werth & Cobia, 1995 [22]	- Ethics of care: “principles alone do not provide a comprehensive basis for the most important ethical decisions”. - Slippery slope. - Autonomy versus beneficence principles. - Meaning of a caring relationship (exploration of feelings – including caregivers’; meaningful communication; thoughtful decision making). - Lack of guidelines from AMA and ANA.
				**Opinion Studies Specifically Focused on Late Life Rational Suicide**		
**Author, Year**	**Perspective/Approach**	**Sample Features/ Specialists**	**Rating Scales**	**Topics Covered/Reasons Mentioned for RS**	**RS Criteria and/or Definition**	**Main Results and Conclusions**
Conwell & Caine 1991 [38]	Critical position	Psychiatrists Psychogeriatricians Researchers Consultants	n.a.	Ageism Burden Physical illness QoL	n.s.	- Poor attention paid to the effects of psychiatric illness on rational decision making in the context of the debate on RS. - Personal biases possibly affecting the determination of a suicidal person’s “rationality”: about aging, old age, psychological effects of chronic disease. - Suicide in the absence of treatable affective illness is uncommon; critical depressive illness precludes rational decision making. - Differential diagnosis: depressed mood versus sadness developing as a natural response to serious illness. - Peculiar presentation of major depressive illness in old age, reduced use of mental health services on behalf of elderly.
Gallagher-Thompson &Osgood 1997 [42]	Overview of epidemiology of late life S, demographics and risk factors, assessment, RS.	Healthcare professionals	BHS MSSI BDI GDS SCID	Autonomy (loss of) Control (loss of) Dignity (loss of) Disability Hopelessness Loss of meaning in life Losses Pain Poor self-esteem QoL Terminal illness	Diekstra, 1986 [24] Motto, 1972 [23] Battin, 1991 [43] Humphry, 1992 [25] Werth & Cobia, 1995 [22]	- Risk factors for old age S: > 60 y.o., Caucasian, divorced/widowed, no longer employed, poor health, depressed or not, alcohol, access to gun, reduced self-esteem, history of mental illness, history of S, poor relationships. - Arguments in favour: philosophy, autonomy, meaning in life. - Proposed interventions to reduce SI and increase QoL: medication, ECT, support groups.
Humphry 1992 [25]	Position of the leader of the National Hemlock Society, mentioning euthanasia and PAS + case narrative.	n.s.	n.a.	Choice (loss of) Control (loss of) “Living death”Pain/suffering, both physical and emotional Terminal illness	n.s.	- “Suicide and assisted suicide carried out in the face of terminal illness causing unbearable suffering should be ethically and legally acceptable”. - Slippery slope.
Moore 1993 [44]	Historical perspective and discussion of supportive and opposing arguments; implications for nursing. “Old age in our society needs to be viewed once again as a valued status, rather than a cursed disease or a burden”.	Nurses Psychologists	n.a.	Burden Control (loss of) Lack of satisfying role Meaning (loss of)	Weber, 1988b [26] Graber, 1981[27]	- Supportive: right to self-determination; evil of needless suffering; Battin’s 17 considerations for assessment. - Opposing: ageism; slippery slope. - Having no alternative but suicide raises doubts about the rationality of older adults’ decision. - Individual life histories, not aging, are critical for the understanding of suicide in later years.
Prado 2015 [45]	Philosophical and bioethical perspective; discussion of the author’s position.	n.s.	n.a.	Conditions diminishing the individual as a person, irremediable Dependence Hopeless medical situations Irreversible deterioration Pain	n.s.	- “Life is not itself an unconditional good”, and sheer organic survival is not an absolute value. - “Whatever the condition of our bodies, once our minds deteriorate beyond a certain point, we cease to exist as the person we are”. - Proposed objections to S/SA: moral, religious, cultural, social, legal.
Richards 2016 [46]	Empirical/theoretical overview to synthesize knowledge, including existential questions about the perception of complete life or tiredness of life.	n.s.	n.a.	Burden Control (loss of) Dependence Disability, functional Illness, chronic Loneliness Loss of meaning/purpose in life Losses Pain, suffering Personality and coping strategies Psychological issues (depression, cognitive decline) QoL Social isolation Tiredness of life	Werth, 1999 [41] Battin, 1991 [43] McCue & Balasubramaniam, 2017 [47]	- Not all SI or planning for S should be unquestioningly pathologized. - Decision making is not a purely cognitive process. - Not all motivating factors for old age RS are open to be remedied. - Importance of end of life care context in which older people find themselves.
Ruckenbauer et al. 2007 [48]	Critical revision of RS. Tension between medical care for patients and patients’ autonomy.	Physicians	n.a.	Burden Conflict Illness Loss Pain Traditional family structures falling apart	n.s.	- Suicide as symptom of individually and/or socially conditioned lack of freedom, rather than of sovereign self-determination. - Underestimation of depression and suicidal potential in old age. - Cult of youthfulness versus old age, associated with weakness, deficiency, increased health costs.
				**Opinion studies not Specifically Focused on Late Life Rational Suicide (but Mentioning it in the Text)**		
**Author, Year**	**Perspective/Approach**	**Sample Features/Specialists**	**Rating Scales**	**Topics Covered/Reasons Mentioned for RS**	**RS Criteria and/or Definition**	**Main Results and Conclusions**
Battin 1991 [43]	S not interpreted as evidence of depression or mental illness; meaning and motivation.	Mental health professionals	n.a.	Terminal illness Disability, severe and permanent Advanced old age	Motto, 1972 (mentioned) [23]	- Presentation and discussion of 17 reasons which should be explored to understand whether S would be rational or not. - “[…] respectful and humane way” to approach “persons who, in a society now beginning to consider S as a rational and even responsible way of avoiding the degradations of terminal illness, severe permanent disability, or extreme old age, wish to explore this option with a trained and insightful professional”.
Clark 1992 [49]	Overview of S and terminal illness, PAS, RTD and euthanasia mentioned; Hemlock Society mentioned. Critical position versus the “understandable reasons” for contemplating S.	Mental health professionals General internists Family physicians	n.a.	Dependence Deteriorating health Disability Helplessness Hopelessness Isolation Loneliness Outliving family members Pain Poverty Severe illness	n.s.	- The so called “understandable reasons” for S rarely stand alone, with no coexisting psychiatric illness, as causes of suicidal thinking. - Almost all persons who die by S evidence symptoms of major psychiatric illness. - Features of depressive illness often overlooked. - Possibility of RS not precluded, but there is likely a strong cultural bias to overlook the “forces and motives implicated in cases of S by older persons”. - The question of mental competence to opt for S.
Siegel 1982 [21]	Evolving societal values concerning death and S, RTD, RTS; Hemlock Society mentioned.	Clinicians	n.a.	QoL Pain Loss of meaning in life Lack of support Self-determination Control (loss of)	Hoche’s “Balance sheet suicide”	- Living will. - Conflictual and ambivalent nature of SA. - Intervention appropriate if (1) the individual is not completely resolved in the decision to die (conflict, ambivalence); (2) the individual does not seem to be realistically appraising his/her problems or prospects for the future.

Legend: AMA: American Medical Association; ANA: American Nursing Association; BDI: Beck Depression Inventory; BHS: Beck Hopelessness Scale; B-SIS: Beck Scale for Suicidal Ideation; B-SLS: Beck Suicide Lethality Scale; CIRS-G: Cumulative Illness Rating Scale adapted for Geriatrics; DDRS: Desire for Death Rating Scale; GDS: Geriatric Depression Scale; HADS: Hospital Anxiety and Depression Scale; HAM-D: Hamilton Depression Rating Scale; MCAS: Montreal Cognitive Assessment Scale; MCAT: Montreal Cognitive Assessment Test; MMSE: Mini-Mental State Examination; MSSI: Modified Scale for Suicidal Ideation; n.a.: not applicable; n.s.: not specified; PAQ: Psychological Autonomy Questionnaire; PAS: Physician Assisted Suicide; QoL: Quality of Life; RS: Rational Suicide; RTD: Right to Die; RTS: Right to Suicide; S: suicide; SA: suicide attempt/suicide attempters; SATHD: Schedule of Attitudes Towards Hastened Death; SCID: Structured Clinical Interview for DSM; SI: suicidal ideation/individuals with suicidal ideation; SIS: Suicide Intent Scale; SPSI-R: Social Problem Solving Inventory-Revised – Short Version; SUD: Substance Use Disorder; VSED: voluntary stopping of eating and drinking; y.o.: years old.

**Table 3 medicina-55-00656-t003:** Arguments in favour and opposing rational suicide.

Arguments in Favour of Rational Suicide	Arguments Opposing Rational Suicide
Moral right to self-determination [33]	Should death wishes, and ideation and action aimed at deliberately ending one’s life ever be considered as “rational”?
Needless suffering [33]	Ageism: old age individuals as a burden; death as a solution for insoluble age-related suffering [12].
Exerting control over one’s death: satisfaction and empowerment.	Slippery slope: from right to die to social obligation to die [9].
Suicidal ideation and behaviour may be the logical and understandable outcome of a balance sheet where death becomes preferable to life [33].	Is suicide *per se* an evidence of mental instability? [33] Having suicidal ideation is often the very reason why an individual is classified as having a mental illness.
Suicide can be a serious and legitimate answer to the individual’s existential situation, which should not be dismissed as a depressive symptom [20]. A history of mood disorder (as well as of any other mental illness) does not mean that the individual’s decision-making capacity is impaired and should be questioned forever [50,51].	Suicide itself is an emotional condition precluding the possibility of rationality: the suicidal individual is usually not capable to consider other option than suicide to a condition of perceived intolerable misery. One would rather live, if a better solution than suicide was at hand [22,52,53,54,55].

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
