# Peer review of "Rational Suicide in Late Life: A Systematic Review of the Literature"

_medicina, 2019, doi:10.3390/medicina55100656_

Round 1

Reviewer 1 Report

Mario Garrett

Review of the paper Rational suicide in late life: a systematic review of the literature by Carla Gramaglia, Raffaella Calati & Patrizia Zeppegno.

This topic is highly pertinent. The authors accomplish in promoting this topic and contributing to the knowledge base. There are some edits that are required to enhance the quality of their argument.

An overview of the topic and the authors’ approach.

The paper addresses the conflicting definitions of what rational suicide means in the literature. If rational suicide exists (i.e., it is rational) then the nosology (how we categorize diseases) needs to change to include death as a rational outcome (with or without medical treatment.) The confound with depression either makes this a problem or else it means that depression itself is a reflection of the ambivalence we feel between the rationality of death and our ambition to deny its eventuality. Depression might be a realization that our ambition to deny death is irrational.

The “slippery slope” or “thin end of a wedge” argument needs to be dispelled by PAS statistics that show that a very small and homogenous group of people avail themselves of this exit strategy. Statistics show there is no great influx of people utilizing PAS but rather a very small and steady number of people over two decades of PAS accessibility statistics.

Also the authors need to address that “ageism” in some of the literature is used inappropriately. The historical argument of eugenics and termination of life for “unwanted” types of people is an important one. But this is self-selected and therefore not imposed by any eugenics motivation.

Specific Comments

L30-32 “Older adults do not only account for a disproportionately high number of suicide deaths, but they are more likely to die by suicide rather than to attempt suicide, compared to younger adults.”

Rephrase to: “Older adults account for a disproportionately high number of suicide deaths because they are more successful at committing suicide compare to younger adults”

L33-35 “In old age there is a high risk of unrecognized and/or untreated psychiatric illnesses (6, 7), and affective illness, particularly major depression, is the most common disorder and the most important risk factor associated with late life suicide (3,8–10).”

Rephrase to: “In older age there is a high risk of unrecognized and untreated psychiatric illnesses (6, 7). In particularly depression is the most common disorder and the most important risk factor associated with late life suicide (3,8–10).”

L44 reference 14 is wrongly cited in the references

L56-57 “…medical knowledge and technology, which have allowed a rapid increase of life expectancy and longevity,…” in fact good public health, clean water, accessible fresh food and good sewage resulted in an increase of life expectancy. Medical intervention has very little influence on aggregate life expectancy.

L77-79 “…kill oneself by nonviolent means (such as self-starvation or noncompliance with essential medical treatment), and by the entanglement of medical, clinical, personal, cultural, ethical, religious and historical issues (28).”

One cannot kill oneself by “entanglement of medical, clinical, personal, cultural, ethical, religious and historical issues” however obtuse the jargon used in those disciplines. The possible meaning is to rephrase the sentence to: “…kill oneself by nonviolent means such as refusal of food and liquids or noncompliance with essential medical treatment. Such means of promoting one’s death causes ambiguity in medical, clinical, personal, cultural, ethical, religious and historical interpretations (28).”

Such interpretation is also faithful to the reference.

L81-82: “… (PAD) and physician assisted suicide (PAS) (i.e., if suicide can be rational, then the right-to-die should be present and regulated).”

Change the word present to legal. Rephrased to: “… (PAD) and physician assisted suicide (PAS) (i.e., if suicide can be rational, then the right-to-die should be legal and regulated).”

L175-6 “…and unresolved question is whether suicidal ideation and attempt should be treated as symptoms of a psychiatric disorder or not.”

Yes it is. Our current nosology includes suicidal ideation and suicide attempts as a symptom of either major depressive episodes or borderline personality disorder.

The definition of suicidal behavior in DSM-5 Section III was approved by the Institute of Medicine in 2002 and is consistent with the U.S. Centers for Disease Control and Prevention (CDC) definition and the U.S. Food and Drug Administration definition.

References

Reference 1 in L328 should be: WHO. Suicide in the Western Pacific [Internet]. [Accessed 2019 Aug 16]. Available from: https://www.who.int/westernpacific/health-topics/suicide 329

Reference 14 should be: Erikson EH. Childhood and society. WW Norton & Company; 1993 Sep 17.

Reviewer 2 Report

This manuscript provides a well-organized review of literature on rational suicide involving older adults. An electronic search was conducted of the literature, and using a set of criteria, papers (N=23) were identified for inclusion in the review. Following a brief overview of the topic of rational suicide and definitional criteria of the concept, the results of the reviewed works were discussed in a categorized fashion based on the primary themes and issues that were identified from the various papers. The manuscript concludes that there is no consensus on the topic of rational suicide and its merits in older adults and neither polarized viewpoint seems sufficient to eliminate the other. Thus, additional consideration and investigation of the topic are warranted.

There are some possible issues that might be considered to improve the contribution of the manuscript to the body of knowledge on the topic of rational suicide in late life.

(1) The final paragraph preceding the section “5. Conclusion” (lines 307-309) contains a sentence that highlights what could be a possible improvement. “However, it should be underscored that a limitation of this work is that, since the available literature was very uneven and mainly based on opinions and theoretical papers, the reported quantitative synthesis is limited.” In fact, the manuscript labels the 23 included papers all as “studies” when several of them (perhaps as many as one-third at least) represent no empirical data based on the descriptions in the useful and informative Table 2. One suggestion would be to reorganize that table into subsections such as “Opinions/Theoretical/Position Papers,” “Case Study Methodology/Inclusion,” and “Empirical Studies” (quantitative or qualitative= “studies”). The papers (not all “studies”) each provide a potential contribution to the literature, thinking about as well as practice regarding possible rational suicide thoughts or actions encountered in older adults.

(2) In a related issue, the exclusion criteria include “3) they were book chapters, commentaries and editorials/letters to the editor”. While letters are more understandable given the stated objectives, the exclusion of book chapters (and for that matter even some books, see below) as well as commentaries and editorials seem less defensible with the acceptance/inclusion by the authors of the papers in the final set of 23 that were not empirical, but rather providing opinions, positions, and theoretical arguments (e.g., Battin; Clark; Humphrey; etc.). The argument by the authors is that this review represents “the first attempt to revise (sic?) the literature focused on rational suicide in older adults.” (lines 306-307; the word “revise” appears to be a typo such that the word “review” would seem more appropriate). Excluding earlier books, book chapters, and possibly commentaries or editorials might provide an impression that no review of the issues has previously been published. The particular set of empirical work presented here coupled with the arguments on each side of the “debate” about rational suicide in late life does represent a unique contribution to the discussion of this important issue. The inclusion of earlier works, particularly in books and book chapters, would provide a more complete historical review of the literature. That is, there have been several published works that have addressed rational suicide generally and many have included at least some if not primary focus on older adults. For instance, Prado (1990) published a book with the subtitle of “Preemptive suicide in advanced age,” Portwood (1978) wrote “Common-sense suicide: The final right,” Werth (1999) edited a book “Contemporary perspectives on rational suicide” (in which pairs of professionals in a large variety of fields provided pro and con arguments on rational suicide–one of the pairs were gerontologists specifically) as well as Werth (2016) “Rational suicide? Implications for mental health professionals,” and finally, Battin (1995) and Battin and Mayo (1980) wrote more general books focused on ethical and philosophical issues in suicide with specific discussions of rational suicide included. Certainly all of these and perhaps others need not be included in the review and discussion, but to suggest that the only papers or works that provide important contributions to the discussion and consideration of rational suicide appear in professional journals and these 23 papers seems misleading.

(3) The inclusion of the concept of “silent suicide” among older adults is a valid one for the discussion of rational suicide, but that term is not widely used and the concept has been in the suicidology literature for several decades. For instance, Farberow (1980) edited a book on Indirect Self-Destructive Behavior (ISDB) and Shneidman (e.g., 1973) discussed the concept of “subintentioned death” that would include these behaviors.

Battin, M. P. (1995). Ethical issues in suicide. Englewood Cliffs, NJ: Prentice Hall.

Battin, M. P., & Mayo, D. J. (Eds.). (1980). Suicide: Philosophical issues. New York: St. Martin’s Press.

Farberow, N. L. (Ed.). (1980). The many faces of suicide: Indirect self-destructive behavior. New York: McGraw-Hill

Portwood, D. (1978). Common-sense suicide: The final right. New York: Dodd, Mead.

Prado, C. G. (1990). The last choice: Preemptive suicide in advanced age. New York: Greenwood Press.

Shneidman, E. S. (1973). Deaths of man. New York: Quadrangle.

Werth, J. L., Jr. (Ed.). (1999). Contemporary perspectives on rational suicide. Philadelphia: Brunner/Mazel.

Werth, J. L., Jr. (2016). Rational suicide? Implications for mental health professionals. New York: Routledge.
